# COMPOSITIONAL DISCRETE DIFFUSION FOR IMBALANCED 3D SCENE SYNTHESIS AND DATASET GENERATION

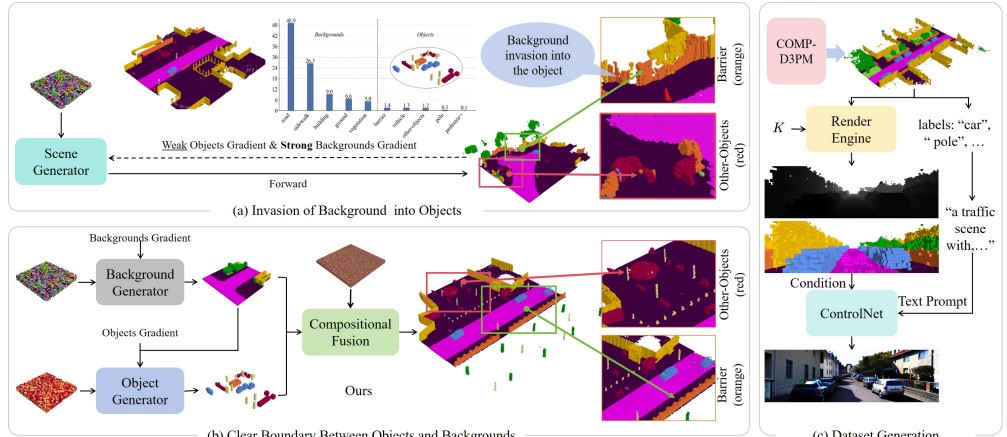

Figure 1: Motivation for Comp-D3PM. (a) The background invasion into objects observed in previous methods. (b)Our Compositional framework that disentangles the generation of background and foreground elements. (c) The pipeline for dataset generation ($K$ is the default projection matrix of the camera). In this process, we employ ControlNet Zhang et al. (2023) to generate images.

## ABSTRACT

3D semantic scene synthesis using discrete diffusion models faces severe challenges due to extreme class imbalance, where background voxels vastly outnumber foreground objects. This imbalance becomes particularly problematic in discrete diffusion for two reasons: (1) the denoising process operates in probability space rather than feature space, making minority classes vulnerable to majority absorption, and (2) learned transition probabilities exhibit systematic bias toward backgrounds, which compounds across diffusion steps, causing irreversible loss of foreground information. We identify this phenomenon as *probabilistic flow collapse*—a fundamental limitation of existing methods. To address this, we propose the Compositional Discrete Denoising Diffusion Probabilistic Model (Comp-D3PM), which synthesizes 3D scenes by compositionally denoising foreground and background voxels through separate transition dynamics. Our contributions are threefold: (1) we formally characterize probabilistic flow collapse and introduce a two-stream architecture that prevents minority-class absorption through compositional modeling; (2) based on this architecture,we enable arange of applications, including the generation of image–semantic scene datasets; and (3) we demonstrate on CarlaSC and SemanticKITTI that Comp-D3PM produces significantly more realistic and diverse scenes while preserving semantic integrity.

## 1 INTRODUCTION

3D semantic scenes provide richer environmental representations than traditional bounding boxes or sparse points, making them essential for autonomous driving and robotics perception Yao et al.

(2023); Huang et al. (2023); Cao & De Charette (2022). However, annotating such scenes is prohibitively expensive and labor-intensive, severely limiting the scalability of supervised approaches. While generative models offer a promising alternative for synthesizing 3D scenes from limited data, previous dataset generation approaches have primarily focused on 2D data Nguyen et al. (2024); Zhang et al. (2021).

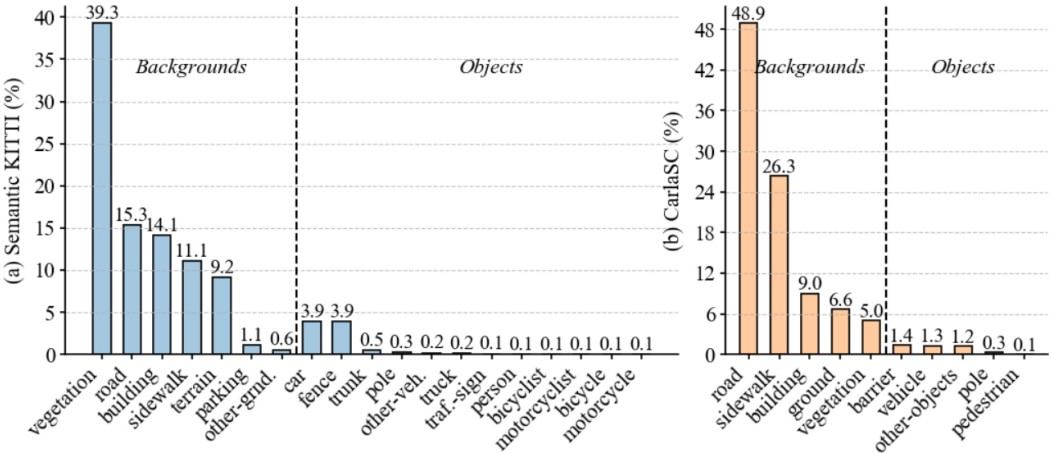

Figure 2: Percentage distribution of classes in the SemanticKITTI Behley et al. (2019) (a) and CarlaSC Wilson et al. (2022) (b) datasets.

Recent advances Lee et al. (2024); Liu et al. (2024); Li et al. (2025); Zhang et al. (2024) have made significant progress in semantic scene synthesis. However, these approaches often overlook a critical challenge: the extreme voxel distribution imbalance between foreground objects and background. This imbalance induces gradient asymmetry, causing models to favor background over foreground, as illustrated in Fig. 3 (right), which in turn exacerbates background invasion into objects, as shown in Fig. 1 (a). This issue is particularly severe in discrete diffusion models, where the denoising process operates in probability space rather than feature space, and any tendency of diffusion steps to favor background classes is further amplified through repeated steps in probability space, resulting in irreversible loss of foreground information. We term this phenomenon probabilistic flow collapse, a fundamental limitation of existing methods.

In this paper, we propose Compositional Discrete Denoising Diffusion Probabilistic Models (*Comp-D3PM*), a framework that addresses the probabilistic flow collapse in discrete diffusion models through principled decomposition of transition dynamics. Building on theoretical insights into discrete state space modeling, Comp-D3PM separates the scene generation process into three progressive stages: 1) generating a low-resolution background to capture the overall scene layout while minimizing interference with potential foreground regions; 2) generating foreground objects conditioned on the background, ensuring that dynamic entities respect background boundaries and maintain fine details;3) jointly refining both components through a compositional fusion network that preserves the learned distributions while ensuring spatial coherence. By effectively resolving gradient asymmetry and preventing background invasion, Comp-D3PM establishes a new benchmark for realistic and semantically coherent 3D scene synthesis, highlighting its significance for future research in structured scene generation.

Our contributions are summarized as follows:

- We identify and formalize the *probabilistic flow collapse* problem in discrete diffusion models for 3D scene generation, showing how extreme voxel distribution imbalances fundamentally disrupt object learning, leading to systematic background invasion artifacts.

- We propose Comp-D3PM, a principled framework that addresses this issue by learning separate transition dynamics for foreground and background, preserving the integrity of each distribution's denoising process while enabling flexible compositional generation.

- We demonstrate Comp-D3PM's practical impact by developing an automated SSC dataset generation pipeline and other applications, constructing and validating a monocular SSC dataset that enables training without manual annotation, showcasing the potential for scalable 3D scene understanding.

- We achieve state-of-the-art performance on CarlaSC Wilson et al. (2022) and SemanticKITTI Behley et al. (2019); Geiger et al. (2013), with significant improvements in both generation quality and physical plausibility, validating our theoretical insights with empirical results.

## 2 RELATED WORK

### 2.1 DIFFUSION MODELS

Diffusion models have recently achieved great success in 2D image synthesis Ho et al. (2020); Nichol & Dhariwal (2021); Dhariwal & Nichol (2021), generating high-quality results Rombach et al. (2022); Peebles & Xie (2023) and enabling diverse tasks such as text-to-image generation Ramesh et al. (2022); Kim et al. (2022) and prompt-based creation Zhang et al. (2023).

Based on iterative denoising, these models produce realistic outputs but at high computational cost. Recent efforts Rombach et al. (2022); Nichol & Dhariwal (2021); Song et al. (2020) reduce this burden, yet most work remains limited to continuous 2D data, with 3D and categorical generation still underexplored.

Our work focus on generating 3D semantic scenes Behley et al. (2019); Tian et al. (2024); Tong et al. (2023), discrete voxelized scenes with semantic labels.

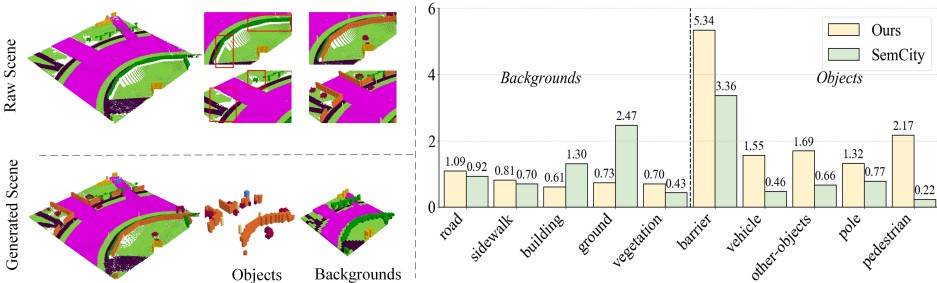

Figure 3: Visualization of decoupling and downsampling. **Raw Scene** shows background decoupling without downsampling and resulting **holes** (left) and the normalized class distribution in 10,000 generated CarlaSC samples with a comparison to the baseline Lee et al. (2024) (right).

### 2.2 CATEGORICAL DATA GENERATION

Generating categorical data is challenging due to the non-differentiable argmax and unordered categories. Traditional continuous-data methods don't apply directly, so a common solution is mapping categories to a continuous latent space for differentiation, then converting back to discrete labels Lee et al. (2024); Regol & Coates (2023).

Another approach involves using techniques like Gumbel-Softmax Hoogeboom et al. (2021) or leveraging Markov transition matrices Austin et al. (2021); Lee et al. (2023). Alternatively, categorical data can be generated alongside its corresponding image, treating the task as a segmentation problem Zhang et al. (2021); Nguyen et al. (2024).

### 2.3 3D DATA GENERATION

Existing methods for 3D generation primarily focus on generating individual objects Vahdat et al. (2022); Luo & Hu (2021), point clouds Caccia et al. (2019); Shu et al. (2019); Yang et al. (2019);

Ran et al. (2024), and rendering-based scene representationsTang et al. (2024); Paschalidou et al. (2021); Gege Gao (2024), while the generation of semantic scenes remains a relatively new area.

For semantic scenes, recent works such as the two-stage three-plane approach Lee et al. (2024) and the three-stage super-resolution approach Liu et al. (2024) have shown promising results. Despite these advances, these methods treat the scene as a whole, overlooking the probabilistic flow collapse. This neglect can cause background invasion into foreground regions, eroding fine object details and limiting the realism of generated semantic scenes.

Additional methods like Zhang et al. (2024) generate scenes conditioned on BEV representations or jointly generate point clouds and images Li et al. (2025), but both rely on BEV distributions compressed from real scenes as conditions. Other methods, such as Bian et al. (2025), attempt to extend semantic scene generation to four dimensions, while Wei et al. (2024) explores integrating scene generation with language models.

## 2.4 DATASET GENERATION

Current dataset generation efforts mainly focus on 2D domains, using GANs Zhang et al. (2021); Li et al. (2022) or diffusion models Nguyen et al. (2024); Wu et al. (2023) to generate data for tasks such as image classification Azizi et al. (2023); Sarıyıldız et al. (2023) and segmentation Zhang et al. (2021); Nguyen et al. (2024). In this work, we present a simple semantic scene-to-image dataset generation framework based on the proposed Comp-D3PM, and validate it through SSC experiments, offering a practical foundation for future research.

## 3 PRELIMINARIES

### 3.1 D3PM BACKGROUND

D3PM Austin et al. (2021) generates structured discrete data like semantic scenes and text via a denoising process. Unlike Gaussian noise in traditional diffusion models Ho et al. (2020), D3PM uses discrete-specific noise applied through transition matrices $\mathbf{Q}_t$. For a discrete variable $x_t$ with $K$ categories, the forward process at step $t$ given $x_0$ is described by the cumulative transition matrix $\bar{\mathbf{Q}}_t = \mathbf{Q}_0 \mathbf{Q}_1 \ldots \mathbf{Q}_t$:

$$q(x_t|x_0) = \text{Cat}(x_t; p = x_0 \bar{\mathbf{Q}}_t) \tag{1}$$

where $x_0$ is represented by the one-hot row vector, and $\text{Cat}(x; p)$ is a categorical distribution over $x$ with the probabilities given by the row vector $p$.

Since the transition matrix $\mathbf{Q}_t$ is bidirectional, irreducible, and aperiodic, $x_t$ converges to a stationary distribution as $t$ grows. The reverse process uses a model with parameters $\theta$ to predict $p_\theta(x_0|x_t)$, approximating $x_0$ from the true posterior $q(x_{t-1}|x_t, x_0)$, which can be expressed as:

$$q(x_{t-1}|x_t, x_0) = \text{Cat}\left(x_{t-1}; p = \frac{x_t \mathbf{Q}_t^T \odot x_0 \bar{\mathbf{Q}}_{t-1}}{x_0 \bar{\mathbf{Q}}_t x_t^T}\right) \tag{2}$$

where the denominator $x_0 \bar{\mathbf{Q}}_t x_t^T$ is a normalization term ensuring that the probabilities sum to 1.

### 3.2 PROBABILISTIC FLOW COLLAPSE IN IMBALANCED DISTRIBUTIONS

While D3PM performs well on balanced categorical distributions, severe class imbalances in 3D semantic scenes give rise to a phenomenon we term *probabilistic flow collapse*. In typical scenes, background voxels constitute over 90% of the volume (95.8% in CarlaSC Wilson et al. (2022) and 90.7% in SemanticKITTI Behley et al. (2019)), leaving foreground objects with less than 10%. This extreme imbalance fundamentally distorts the learned transition dynamics. In D3PM, the reverse process relies on approximating the posterior $q(x_{t-1}|x_t, x_0)$ through the model's prediction $p_\theta(x_0|x_t)$. Since the training objective is dominated by background voxels—which appear in over 90% of positions—the model learns to preferentially predict background classes. This manifests as systematic bias in the effective transition probabilities:

$$P(x_{t-1} = j|x_t = i) \propto \sum_{x_0} p_\theta(x_0|x_t = i)q(x_{t-1} = j|x_t = i, x_0) \tag{3}$$

Due to the imbalanced training data, these transition probabilities exhibit strong asymmetry:

$$P(x_{t-1} = j|x_t = i) \approx \begin{cases} \alpha_t & \text{if } j \in B \text{ (background)} \\ \beta_t & \text{if } j \in F \text{ (foreground)} \end{cases} \qquad (4)$$

where $\alpha_t \gg \beta_t$ regardless of the current state $i$. This bias compounds over diffusion steps. After $T$ steps, the cumulative probability of transitioning from any foreground class to background becomes:

$$\sum_{j \in \mathcal{B}} P(x_0 = j|x_T \in F) \approx 1 - \epsilon, \quad \epsilon \to 0 \text{ as } T \to \infty \qquad (5)$$

This represents an irreversible collapse: once foreground information flows into background states, the reverse process cannot recover it, as the model has learned to always predict background as the most likely original state.

**Consequences of Probabilistic Flow Collapse:**

1. **Irreversible Information Loss**: Once foreground voxels transition to background, the reverse process cannot recover them.

2. **Background Invasion**: During generation, the model preferentially samples background categories even in object regions, eroding boundaries and fine details, as shown in Fig.1(a).

3. **Mode Collapse for Rare Classes**: Classes with minimal voxel counts (e.g., motorcycles, pedestrians) disappear entirely as their Probabilistic Flow collapse into dominant backgrounds, as shown in Fig.3(right).

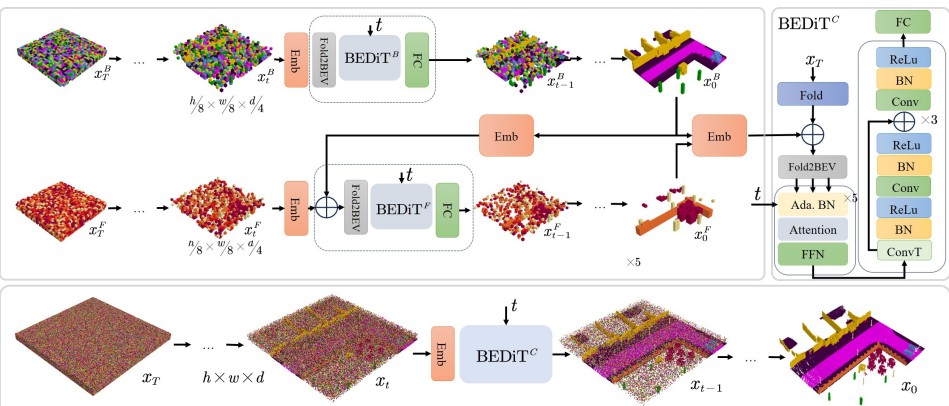

Figure 4: Framework of the proposed Compositional discrete diffusion Model.

## 4 METHOD

In this section, we introduce Compositional D3PM (Comp-D3PM) (see Fig. 4), a framework specifically designed to address severe gradient imbalance between foreground and background and to suppress background invasion into foreground objects. At the same time, we introduce a series of controlled generation tasks, such as inpainting and dataset generation.

### 4.1 OBJECT-BACKGROUND DISENTANGLING

To address the gradient imbalance caused by uneven voxel distributions of object and background in semantic scenes, following Kirillov et al. (2019), we separate object and background based on their voxel counts and countability (Fig. 2). This decoupling effectively prevents the gradients of objects with fewer voxels from being overwhelmed, thereby mitigating the background invasion into foreground objects observed in baseline methods (Fig. 1(a)).

However, naively separating voxel categories into object and background distributions often results in noticeable holes in the background, as shown in Fig.3(left). To mitigate this, we combine compositional modeling of categories with downsampling, effectively filling these holes and producing a more coherent and seamless scene

Downsampling categorical data is inherently challenging, as the discrete nature of the data prevents interpolation, leading to potential voxel information loss. To address this, we adopt a score-based downsampling strategy, which preserves scene information while maintaining coherence.

Specifically, for each voxel pool $j$ to be downsampled, if the proportion of empty or invalid voxels exceeds a threshold $th$, the downsampled value is set to either empty or invalid. Otherwise, the pool is assigned the category $k$ with the highest score $s_j^k$, calculated in a TF-IDF (Term Frequency-Inverse Document Frequency) manner, where the count of voxels belonging to category $k$ is normalized by its overall frequency $F^k$ in the dataset:

$$s_j^k = \frac{\text{Count}_j^k}{\sqrt[3]{F^k}} \tag{6}$$

## 4.2 COMPOSITIONAL D3PM

The framework of our model is illustrated in the Fig. 4. We begin by separating the scene into object and background voxel distributions. Each distribution is then downsampled to achieve more coherent and seamless results. To further reduce the number of tokens while preserving maximal information, we introduce BEV-aware DiTPeebles & Xie (2023) (BEDiT), which compresses height-space features and performs attention in the BEV space. Specifically, the diffusion process for both distributions is modeled sequentially: (Emb denotes embedding operation.)

$$\begin{cases} \text{logits}_{t-1}^B = \text{BEDiT}^B \left( \text{Emb}(x_t^B) \right) \\ \text{logits}_{t-1}^F = \text{BEDiT}^F \left( \text{Emb}(x_0^B) \oplus \text{Emb}(x_t^F) \right) \end{cases} \tag{7}$$

For each distribution, we construct a conditional discrete diffusion process. The model for the background distribution $\text{BEDiT}^B$ employs self-conditioning, while the model for the object distribution $\text{BEDiT}^F$ conditions on the background $x_0^B$. This decoupling prevents the gradients of sparse foreground objects from being overwhelmed by the abundant background voxels, allowing the model to learn more accurate and detailed object representations. Finally, the compositional fusion model $\text{BEDiT}^C$ integrates both the background $x_0^B$ and object distributions $x_0^F$ as conditions, focusing on producing a coherent and detailed compositional scene:

$$\text{logits}_{t-1}^C = \text{BEDiT}^C \left( \text{Emb}(x_t) \oplus \text{Emb}([x_0^B, x_0^F]) \right) \tag{8}$$

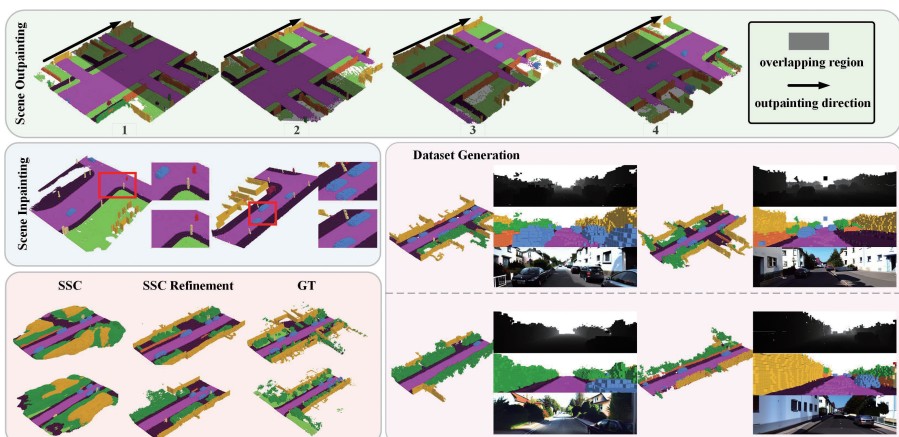

Figure 5: Visualization results of scene manipulation, including scene outpainting, scene inpainting, SSC refinement(using SSC outputs from Cao & De Charette (2022)), and dataset generation.

## 4.3 APPLICATIONS WITH COMP-D3PM

Beyond its generative capabilities, Comp-D3PM is applicable to various tasks, including inpainting and outpainting, semantic scene completion (SSC) refinement, and, most importantly, dataset generation, as illustrated in Fig.5.

### 4.3.1 Dataset Generation

As illustrated in Fig.1 and Fig.5, we leverage the semantic scenes generated by Comp-D3PM to construct a monocular SSC dataset. Specifically, we render each semantic scene onto the image plane using a rendering engine with a default intrinsic matrix, producing paired RGB and depth images. These serve as conditional inputs for training ControlNet Zhang et al. (2023) to synthesize realistic images from 3D layouts. To further enhance semantic alignment, we follow the prompt construction strategy of Nguyen et al. (2024), incorporating object category labels into natural language descriptions. For instance, given scene labels such as "car" and "building", we create prompts like "a traffic scene with: car, building" as textual guidance. Using this pipeline, we construct a new monocular SSC dataset based on SemanticKITTI Behley et al. (2019) and validate its quality by training a representative SSC model on the generated data. This validates the quality of our generated scenes and highlights Comp-D3PM's potential for 3D scene understanding dataset generation.

### 4.3.2 Other Applications

We also employ Comp-D3PM for outpainting, inpainting, and SSC refinement, as illustrated in Fig. 3. However due to space limitations, please refer to the Appendix for details.

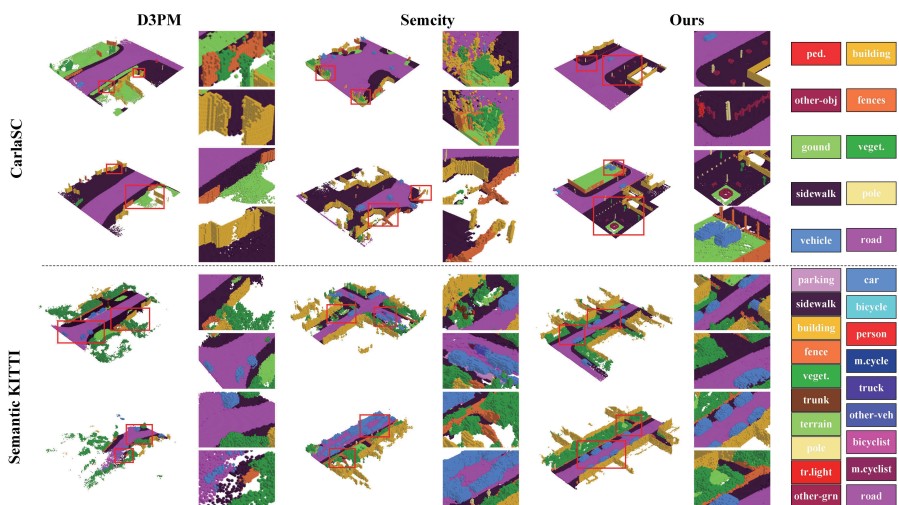

Figure 6: Visualization comparison results of unconditioned generation on the Semantic KITTI Behley et al. (2019) and CarlaSC Wilson et al. (2022) datasets.

## 5 Experiments

### 5.1 Evaluation Protocols

Evaluating the quality of generated 3D scenes is challenging, as common 2D metrics like FID Heusel et al. (2017) and KID Bińkowski et al. (2018) cannot be directly applied. To address this, we adopt four specialized metrics, following the practices in Lee et al. (2024) and Liu et al. (2024).

**F3D**: An adaptation of the *Fréchet Inception Distance* (FID) for 3D data, F3D measures the distance between real and generated scenes in the feature space of a pretrained 3D autoencoder.

**K3D (MMD)**: Similarly, K3D adapts the *Kernel Inception Distance* (KID) to 3D data, computing the *Maximum Mean Discrepancy* (MMD) between features of real and generated scenes.

**2D Metrics**: To leverage existing 2D evaluation methods, we render the 3D scenes into 2D images (Fig.6) and compute FID, KID, and ISCSalimans et al. (2016), similar to Lee et al. (2024).

**Background Invasion Score (BIS)**: To specifically evaluate background invasion into objects in the generated results, we introduced a dedicated human evaluation metric. Five independent raters

blindly scored 300 generated samples on a 0–10 scale, and the mean and variance of these scores were computed to quantify this phenomenon.

Table 1: Comparison with state-of-the-art on Semantic KITTI Behley et al. (2019) dataset. (F3D is measured in units of $10^{-2}$, and B denotes BEV (Bird's Eye View). )

| | F3D↓ | K3D↓ | FID↓ | KID↓ | ISC↑ | BIS↑ |
|---|---|---|---|---|---|---|
| D3PM | 83.2 | 113.0 | 66.0 | 0.0644 | 2.50 | 6.37±0.44 |
| Semcity(B) | 65.7 | 34.3 | 58.5 | 0.0602 | 2.49 | 6.41±1.47 |
| Semcity | 62.3 | 32.4 | 48.2 | 0.0483 | 2.66 | 6.49±2.69 |
| Ours | **53.1** | **26.5** | **24.4** | **0.0179** | **3.32** | **6.86±0.80** |

Table 2: Comparison with state-of-the-art on CarlaSC Wilson et al. (2022) dataset.

| | F3D↓ | K3D↓ | FID↓ | KID↓ | ISC↑ | BIS↑ |
|---|---|---|---|---|---|---|
| D3PM | 96.5 | 30.4 | 59.2 | 0.0551 | 3.06 | 7.91±0.01 |
| Semcity(B) | 87.6 | 29.1 | 45.1 | 0.0370 | 2.96 | 7.78±0.14 |
| Semcity | 80.5 | 16.6 | 26.9 | 0.0178 | 2.72 | 7.88±0.04 |
| PDD | 101.5 | 26.2 | 42.0 | 0.0392 | 3.32 | 7.98±0.12 |
| Ours | **26.3** | **5.6** | **9.8** | **0.0041** | **3.33** | **8.18±0.01** |

## 5.2 EXPERIMENT SETTINGS

We conduct experiments on two datasets: CarlaSC Wilson et al. (2022) and Semantic KITTI Behley et al. (2019). Semantic KITTI is a real-world outdoor dataset based on the KITTI dataset Geiger et al. (2013), containing 20 classes and 3,824 scenes, each represented as a $256 \times 256 \times 32$ occupancy grid. CarlaSC is a synthetic dataset generated from the Carla simulator Dosovitskiy et al. (2017), featuring 32,400 scenes across 10 classes. In this paper, we use the updated version of CarlaSC, where each scene has a resolution of $256 \times 256 \times 16$. All experiments were conducted on an NVIDIA RTX 4090 GPU, and models were optimized using the AdamW optimizer Loshchilov (2017).

We apply BEV folding and downsampling to both objects and background (e.g., reducing a $256 \times 256 \times 32$ voxel grid to $32 \times 32$), where the additional stage only consumes about one-sixth the memory of full-resolution BEV methods, keeping the overall complexity comparable to non-compositional baselines. As previously mentioned, we primarily categorize classes as objects or backgrounds based on their percentage distribution and semantics, as shown in Fig. 2.

## 5.3 COMPARISONS

We mainly compare with semantic scene generation methods that do not require any conditioning Lee et al. (2023; 2024); Liu et al. (2024), as shown in Tab. 1 and Tab. 2. Among them, Semcity Lee et al. (2024) is a discrete diffusion-based method, while Semcity BEV refers to its BEV-based implementation using latent diffusion Rombach et al. (2022). Our method achieves state-of-the-art performance across all metrics, demonstrating its effectiveness in capturing complex scene structures and generating high-quality 3D reconstructions. To ensure a fair comparison, we use the released weights of the involved methods for generation and keep all settings identical across experiments, except for the generation method itself.

Furthermore, Fig. 6 compares our method with others, showing superior detail capture, structural integrity, and accurate shape restoration of small objects. Our approach produces more realistic 3D scenes with sharper boundaries and better spatial arrangement than baselines.

## 5.4 ABLATION STUDIES

**Ablation studies on compositional foreground-background modeling.** We conducted ablation studies on the compositional modeling mechanism by developing a two-stage generation pipeline on the CarlaSC dataset using the same diffusion model for comparison. In the first stage, a down-sampled version of the scene is generated, and in the second stage, this intermediate output is used as

Table 3: Ablation Study on Compositional Modeling Foreground/Background and Downsampling.

| Method | Resolution | F3D↓ | K3D↓ | FID↓ | KID↓ | ISC↑ |
|---|---|---|---|---|---|---|
| Compositional | $h \times w \times d$ | 88.3 | 74.2 | 51.6 | 0.0408 | 3.30 |
| Compositional | $\frac{h}{4} \times \frac{w}{4} \times \frac{d}{2}$ | 46.6 | 12.8 | 18.34 | 0.0162 | 3.32 |
| Compositional | $\frac{h}{16} \times \frac{w}{16} \times \frac{d}{8}$ | 49.5 | 17.9 | 19.25 | 0.0231 | 3.31 |
| Non-compositional | $\frac{h}{8} \times \frac{w}{8} \times \frac{d}{4}$ | 37.6 | 9.85 | 13.87 | 0.0073 | 3.30 |
| Compositional | $\frac{h}{8} \times \frac{w}{8} \times \frac{d}{4}$ | **26.3** | **5.6** | **9.83** | **0.0041** | **3.33** |

a condition to reconstruct the scene at full resolution. As summarized in Tab. 3, compositional modeling of the foreground and background significantly improves the quality of the generated scenes.

To further investigate the necessity of compositional modeling for small object generation, we analyzed the proportion of voxels for each category in the generated samples of the CarlaSC dataset relative to the whole, as shown in Fig. 3(right). Our approach demonstrates a stronger preference for generating objects, while the baseline method Lee et al. (2024), which processes the 3D scene as a single entity, leans more towards generating background components. Additionally, Fig. 6 demonstrates that our approach is capable of accurately generating the shapes of various small objects within the 3D scene, particularly for the *other-object* category, which encompasses a diverse range of shape distributions.

**Ablation studies on downsampling during generation.** As mentioned earlier, performing background-object compositional modeling directly at the original resolution results in hollow regions in the background, as shown in Fig. 3(left). This leads to the leakage of object positional information, as the network can infer the location and category of objects through these hollow regions, thereby reducing the richness of the generated output. At the same time, generating directly at the original resolution also increases the difficulty of network convergence.

We conducted ablation studies on the downsampling process during compositional modeling, as shown in Tab. 3. The method employing full resolution demonstrates significantly inferior results compared to the approach that utilizes downsampling for both background and object generation, highlighting the importance of downsampling.

Table 4: Ablation Analysis for Generated Dataset.

| Method | Real Dataset | | Synthetic Dataset | |
|---|---|---|---|---|
| | IoU↑ | mIoU↑ | IoU↑ | mIoU↑ |
| VisHall3D | 46.14 | 17.06 | 34.33 | 7.17 |
| MonoScene | 37.12 | 11.50 | 24.23 | 4.72 |

**Ablation Analysis for Generated Dataset.** Based on Comp-D3PM, we construct a monocular SSC dataset derived from SemanticKITTIBehley et al. (2019), ensuring no data leakage by training all generative models solely on the training split. To evaluate the quality and utility of the generated dataset, we conduct an ablation study with the state-of-the-art monocular SSC method VisHall3D (ICCV 2025) Lu et al. (2025) and the representative baseline MonoScene (CVPR 2023) Cao & De Charette (2022). Specifically, we evaluate the effectiveness of our generated dataset by comparing zero-shot performance and validating on the original validation set. As shown in Tab. 4, although models trained purely on synthetic data still underperform compared to those trained on real data, the results highlight the promise of our approach as a foundation for future data generation efforts.

# 6 CONCLUSION

In this work, we identify probabilistic flow collapse and propose Comp-D3PM, a two-stream architecture separating foreground and background dynamics. Beyond enhancing generation quality, it adapts to tasks like inpainting, outpainting, and SSC refinement, and enables Comp-D3PM to build a monocular SSC dataset for scalable 3D data generation.

## 7 ETHICS STATEMENT

This research adheres to the ICLR Code of Ethics. No human subjects, personally identifiable information, or sensitive private data were used in this study. All datasets employed are publicly available or released with appropriate licenses, and their use complies with relevant legal and ethical guidelines. The methods proposed in this paper are intended for academic research purposes and do not aim to enable harmful applications. We have taken care to minimize potential biases and discrimination in the data and methods, and we transparently document all procedures to support reproducibility and research integrity.

## 8 REPRODUCIBILITY STATEMENT

We have made efforts to ensure the reproducibility of our results. All experimental setups, model architectures, and training procedures are described in detail in the main text and supplementary materials. The datasets used are publicly available, and data processing steps are documented in the appendix. For the proposed models and algorithms, code will be provided as anonymous supplementary material, enabling reviewers to reproduce the results. Any theoretical claims are supported by detailed derivations included in the appendix. We encourage readers to refer to the main paper, appendix, and supplementary materials for full reproducibility.

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

# A  APPENDIX

## A.1  OTHER APLLICATIONS

### A.1.1  INPAINTING AND OUTPAINTING

Scene inpainting and outpainting are techniques used to enhance and extend generated scenes. With Comp-D3PM, which decouples object and background distributions, scene inpainting is achieved by controlling object distribution generation. As shown in Fig.5, different object distributions can be generated with the same background. Object control is further refined using a mask mechanism; given a known object distribution $x_{known}$, we regenerate specific parts by applying a mask $m$, overriding the input at time $t$ with $\bar{x}_t$:

$$\bar{x}_t = m \otimes x_t + (1 - m) \otimes x_{known} \tag{9}$$

### A.1.2  SSC REFINEMENT

When reconstructing 3D occupancy from visual sensor data Yao et al. (2023); Huang et al. (2023); Cao & De Charette (2022), networks often struggle to restore fine object details, as shown in Fig.5. Previous work Lee et al. (2024) has attempted to refine network-predicted occupancy using generative models with promising results. In our approach, since shape and position are decoupled, we downsample the predicted occupancy, separate object and background distributions, and input them into the upsampling model to reconstruct the shape .

For more visualized inpainting and outpainting samples, please refer to the demo.mp4 in the supplementary material.

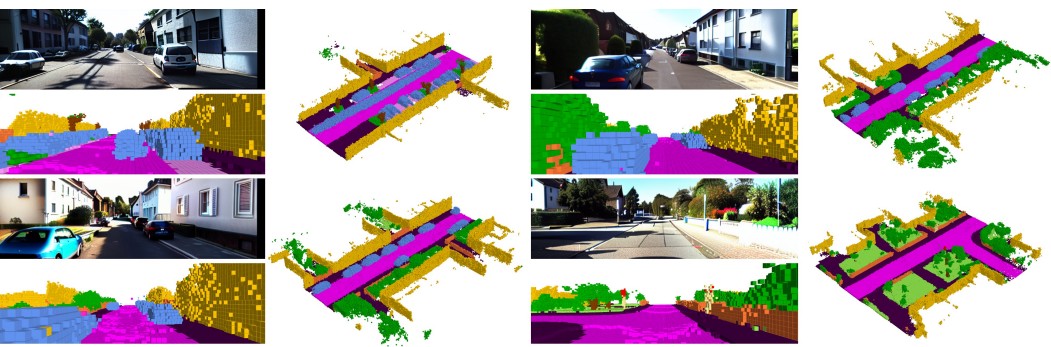

Figure 7: Additional visualization results of the generated dataset.

## A.2 ADDITIONAL RESULTS AND TRAINING DETAILS

To further demonstrate the effectiveness of our approach, we present additional qualitative results on the SemanticKITTI Behley et al. (2019) and CarlaSC Wilson et al. (2022) datasets. As shown in Fig.9 and Fig.8, our method achieves high performance in both scene-level coherence and object-level detail. Notably, our results do not exhibit confusion between different object categories.

In addition to the examples presented in the main text, we also provide an expanded set of generated samples from the OCC-Image dataset in the appendix as shown in Fig.7. These additional samples are intended to give readers a more comprehensive view of the dataset's characteristics, the diversity of the generated outputs, and the effectiveness of our approach, thereby serving as a valuable reference for further research and comparison.

As described in the main paper, we trained Comp-D3PM using AdamW Loshchilov (2017) optimizer on both the SemanticKITTI Behley et al. (2019) and CarlaSC Wilson et al. (2022) datasets for 500 epochs. The batch size was set to 4, and the learning rate was $1 \times 10^{-4}$. The learning rate was decayed by a factor of 10 at the 320th and 420th epochs.

## A.3 LLM USAGE STATEMENT

We confirm that large language models (LLMs) were used solely as a general-purpose writing assistant to improve the clarity, grammar, and readability of the manuscript. The LLM did not contribute to the research ideation, experimental design, data analysis, or interpretation of results, and all scientific content, reasoning, and conclusions are entirely the authors' own work.

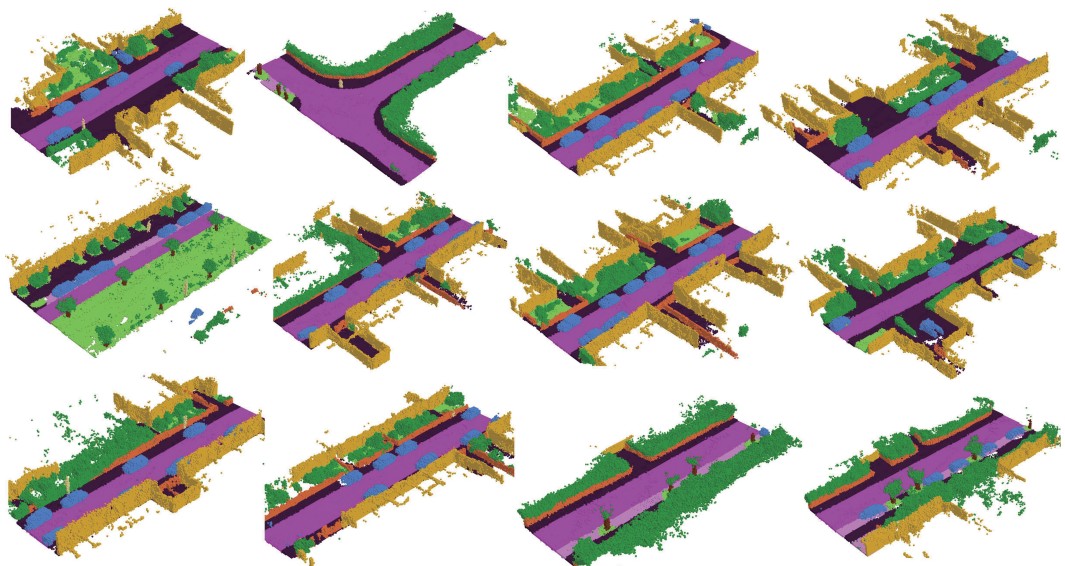

Figure 8: Additional visualization results on the Semantic KITTI dataset.

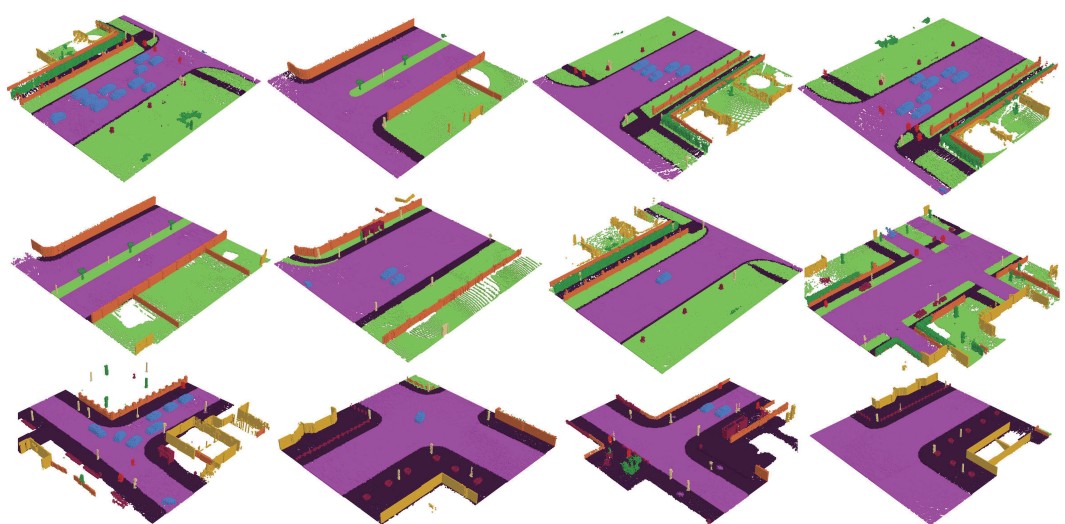

Figure 9: Additional visualization results on the CarlaSC dataset.

