# OpenReview forum: "Compositional Discrete Diffusion for Imbalanced 3D Scene Synthesis and Dataset Generation"
_ICLR.cc/2026/Conference — Submitted to ICLR 2026_

### Official Review · Reviewer_XMZv · 2025-10-26

**Soundness:** 3
**Presentation:** 3
**Contribution:** 3
**Rating:** 4
**Confidence:** 4

**Summary:**

This paper identifies a key problem of class imbalance in 3D semantic scene generation tasks and formalizes this issue with the novel concept of "probabilistic flow collapse." To address this issue, a compositional diffusion model (Comp-D3PM) is proposed, which decouples foreground and background generation, showing a significant improvement compared to baselines. The authors conduct various experiments to illuminate the effectiveness of Comp-D3PM. Ablation studies substantiate the efficacy of the core innovation. The paper is well-motivated, the arguments are sound, and the experimental results are impressive, clearly demonstrating enhanced semantic integrity. Furthermore, the work shows potential for downstream applications such as synthetic dataset generation.

**Strengths:**

1. In response to the identified problem, the Comp-D3PM framework offers an intuitive and effective solution. By decomposing the scene into foreground and background components and learning independent transition dynamics for each, the method directly addresses the root cause of "probabilistic flow collapse." This "divide-and-conquer" strategy is exceptionally clear.

2. The proposed method achieves performance far superior to the existing state-of-the-art (e.g., Semcity) across all core metrics (F3D, K3D, FID, KID) on both real-world (SemanticKITTI) and synthetic (CarlaSC) datasets.

3. The paper does not stop at the generation task itself but also showcases Comp-D3PM's applicability to scene completion, editing, refinement, and, most importantly, "dataset generation."

**Weaknesses:**

1. While the paper's ablation studies validate the effectiveness of "compositional modeling",  they do not provide a dedicated experiment to demonstrate the necessity of the third-stage "BEDiT_C". It is currently unclear how a two-stage model (generating and directly merging the foreground and background) would perform compared to the full three-stage pipeline.

2. The method cleverly addresses the imbalance between foreground and background. However, a severe class imbalance within the foreground often exists (e.g., 'cars' are far more numerous than 'pedestrians'). The paper should discuss whether the "probabilistic flow collapse" phenomenon might also occur within the foreground generation.

3. The paper qualitatively claims that the overall complexity of the three-stage model is "comparable" to baseline methods. Authors should compare the parameter count and inference time with baselines.

4. Although the downsampling strategy effectively resolves the "hole" issue in the background, it is inherently a low-pass filtering operation. This could lead to the loss of high-frequency details, such as sharp object edges and fine geometric structures, resulting in over-smoothed outputs. The paper does not address this potential negative side effect, nor does it analyze whether the employed metrics (like F3D/K3D) are insensitive to such high-frequency detail loss.

5. The generated dataset lacks a quality assessment. It is recommended to calculate the FID and KID metrics against the KITTI dataset.

6. To assess the value of the synthetic data, one could perform hybrid training with real data to demonstrate the resulting performance improvement.

**Questions:**

1. Could you provide an ablation study that compares the full three-stage model with a two-stage version that only generates the foreground and background and then directly merges them? This would clarify the specific performance gain brought by the BEDiT_C fusion module.
2. When calculating metrics, was the Ground Truth downsampled, or was the original data used?

3. Could you please elaborate on the partitioning of the data processing and training stages? Specifically, are the foreground, background, and their fusion trained in a stepwise manner?

4. What is the generalizability of your method, and can it be effectively transferred to related approaches? A discussion on this aspect would further underscore the value and significance of the proposed method.

5. If possible, could you provide further theoretical derivations regarding how class imbalance accumulates across the diffusion steps?

---

> ### Author Response · Authors · 2025-11-19
>
> Dear Reviewer,
>
> Thank you for your valuable feedback.
>
>
> 1. Although we have conducted non-compositional comparison experiments from low-resolution to high-resolution in Table 3 and the ablation studies, we did not include comparisons for generating the complete scene directly from the background.
>
> 2. Class imbalance also exists among objects, but it is not as significant as the imbalance between objects and the background. Moreover, objects are typically not spatially crowded or adjacent to each other (while object and background voxels are often adjacent, voxels between objects are rarely adjacent), so voxel-level interference between objects is relatively minor.
>
> 3. Since the generation of objects and background uses low-resolution BEV representation, the total time required for both is roughly one-third of directly generating a full-resolution scene. With BEDit_c, the overall time is about 4/3 of a single-stage method, while achieving significantly better generation quality.
>
> 4. As discussed in the paper, downsampling effectively decouples object location from shape details (high-frequency details), and high-frequency details are entirely generated by BEDit_c. This reduces the generation difficulty at each stage and improves results. Without downsampling, generated objects and background would have poor details, and using them as high resolution conditions for BEDit_c would make it overly rely on the conditions and prevent BEDit_c from learning to generate high-quality high-frequency details during training.
>
> 5. The metric is computed between the high-resolution scene output by BEDit_c and the high-resolution ground truth (GT).
>
> 6. A semantic scene is a 3D scene composed of voxels with semantic labels. Since the semantic labels are already available, the scene can be partitioned directly according to these labels. During training, the three stages are trained separately, while during sampling, all three stages are used together.
>
> In summary, due to the unsatisfactory score, we will further optimize our work. Please look forward to the next improvements of Comp-D3PM. We wish you all the best.
>
> Sincerely,
> Authors of Comp-D3PM

---

### Official Review · Reviewer_yPUW · 2025-10-29

**Soundness:** 3
**Presentation:** 2
**Contribution:** 2
**Rating:** 4
**Confidence:** 5

**Summary:**

The paper addresses the issue of class imbalance in 3D semantic scene synthesis, where background voxels dominate and foreground details are lost during diffusion. The authors identify this problem as probabilistic flow collapse and propose Comp-D3PM, a compositional discrete diffusion framework that models foreground and background separately before fusing them into coherent scenes. The method incorporates downsampling and BEV-aware architectures to improve training stability and efficiency. Experiments on SemanticKITTI and CarlaSC show performance gains over prior work, and the approach also enables the automatic generation of synthetic 3D datasets for semantic scene completion.

**Strengths:**

1. The paper identifies and formally defines the phenomenon of probabilistic flow collapse in discrete diffusion models, providing a clear theoretical explanation for how class imbalance leads to foreground information loss.

2. The proposed Comp-D3PM introduces a compositional framework that disentangles foreground and background diffusion processes, offering a conceptually simple yet effective way to address class imbalance in 3D scene generation.

3. Beyond scene synthesis, the framework is extended to practical applications such as semantic scene completion, inpainting, and automatic dataset generation, showing good generality and real-world potential.

**Weaknesses:**

1. According to the foreground–background division shown in Figures 1 and 3, vehicles should belong to the foreground. However, in Figure 4, the background generation branch also produces vehicles, which appears inconsistent with the stated compositional setup.

2. In Section 4.1, the paper claims that downsampling resolves background holes, but $BEDiT^C$ later uses the downsampled and fused low-resolution scene as a condition to generate the final high-resolution output. Since the training scenes already contain holes at high resolution, it is unclear how the final results avoid reintroducing them.

3. The PDD framework also performs downsampling and pyramid-based generation. Although its low-resolution representations are hole-free, the final full-resolution outputs still contain holes. It is unclear whether Comp-D3PM effectively overcomes this issue.

4. The proposed score-based downsampling strategy is not experimentally compared to PDD’s approach. The claimed advantages remain theoretical without quantitative evidence.

5. The paper does not report detailed training or inference costs. A direct comparison with SemCity and PDD in terms of GPU memory usage, training time, and inference speed would clarify the computational trade-offs.

6. Since a major goal is reducing background invasion into objects, relying solely on human evaluation (BIS) is insufficient (for evaluating the Background Invasion).

7. While the separation of background and foreground may help scene generation, all examples are unconditional and randomly generated. Without explicit control, the model might overfit, producing scenes overly similar to the training data. Do you have any analysis of this?

8. Typographical and formatting issues:
   - Line 27: missing space in “(b)Our”
   - Figure 1(a): “pedestrian” label partially covered
   - Figure 2: oversized and low resolution
   - Figure 3: unclear axis units and inappropriate placement in Related Work
   - Line 152: missing space between “labels” and citation
   - Line 161: missing spaces after “objects” and “clouds”
   - Line 162: missing space after “representations”
   - Line 286: missing space after “DiT”
   - Lines 332–333: incorrect quotation marks
   - Line 375: missing space after “ISC”
   - Line 471: missing space after “SemanticKITTI”

**Questions:**

Refer to Weaknesses.

---

> ### Author Response · Authors · 2025-11-19
>
> Dear Reviewer,
> Thank you for your valuable feedback.
>
> 1. We apologize for Fig. 4 — the mistake occurred when selecting samples for the figure; the wrong samples were chosen.
> 2. The appearance of holes is due to our decoupling of objects and background. The high-resolution scenes generated by BEDit_c do not decouple objects and background, so they do not contain holes. We use downsampling to remove holes because holes can give away hints about object locations and classes (e.g., seeing a vehicle-shaped hole on the road indicates a car should be there), which could reduce the generalization ability of the generative model.
> 3. The holes in PDD samples are not the same as the “holes” referred to in our paper. In our paper, holes refer to gaps left on the background (e.g., the road) when objects are decoupled from the background. In contrast, the holes in PDD-generated samples arise because the semantic scene GT in the dataset is aggregated from multiple LiDAR frames. Occluded areas cannot be scanned, resulting in holes, and generative models trained on such GT naturally learn this, making these holes unavoidable.
> 4. We did not compare with PDD on SemanticKITTI because PDD’s GitHub does not provide pretrained weights for SemanticKITTI. Training PDD ourselves on SemanticKITTI could lead to controversial results.
> 5. Our method runs at approximately 1.5× the speed of SemCity during sampling, comparable to PDD.
> 6. In addition to human evaluation (BIS) as a metric, we also use 2D and 3D FID/KID and ISC to evaluate generation quality, whereas the baseline methods SemCity and PDD only used one of 2D or 3D metrics.
> 7. Scene generation can be controlled by manipulating the distributions of background and objects as controls.
>
> In summary, due to the unsatisfactory score, we will further optimize our work. Please look forward to the next improvements of Comp-D3PM. We wish you all the best.
>
> Sincerely, Authors of Comp-D3PM

---

### Official Review · Reviewer_fY8p · 2025-10-30

**Soundness:** 3
**Presentation:** 3
**Contribution:** 2
**Rating:** 4
**Confidence:** 3

**Summary:**

This paper studies the probabilistic flow collapse problem in discrete diffusion models for 3D semantic scene synthesis, where foreground information is absorbed by dominant background classes. The authors propose Comp-D3PM, a compositional two-stream diffusion framework that disentangles foreground and background dynamics to mitigate this issue. The method achieves significant improvements over prior works on SemanticKITTI and CarlaSC, and further supports applications such as SSC refinement and automatic dataset generation.

**Strengths:**

- The paper identifies and formalizes the phenomenon of probabilistic flow collapse in discrete diffusion, providing a clear explanation for failure cases observed in prior 3D scene generation models.
- It proposes a targeted two-stream architecture that disentangles foreground and background diffusion and integrates them through a BEV-aware Transformer (BEDiT), resulting in a well-motivated and coherent design.
- The method achieves competitive results on SemanticKITTI and CarlaSC, showing clear improvements in structural fidelity and semantic consistency over previous approaches.

**Weaknesses:**

- The method is only evaluated on street-level, vehicle-mounted datasets (SemanticKITTI and CarlaSC), leaving its generality to other types of scenes unclear and possibly tied to this specific inductive bias.
- The generated scenes remain relatively simple and lack the structural and semantic complexity observed in real-world 3D environments.
- The paper lacks a comparison or discussion of alternative imbalance-handling strategies, like loss re-weighting

**Questions:**

Have the authors considered applying the proposed compositional discrete diffusion framework to indoor scene generation tasks? It would be interesting to see whether the same idea generalizes beyond driving scenes and whether probabilistic flow collapse also appears in more structured indoor environments.

---

> ### Author Response · Authors · 2025-11-19
>
> Dear Reviewer,
>
> Thank you for your valuable feedback.
>
> 1. We conducted experiments on the SemanticKITTI and CarlaSC datasets because the baseline methods, both conditioned and unconditioned, were evaluated on these two datasets. We aimed to maintain consistency with prior work.
> 2. SemanticKITTI is a real-world dataset, directly derived from real driving scenes.
> 3. The fundamental reason why decoupling objects and background is effective is that it allows the generative model to learn a more structured distribution. This enables the model to focus on learning the distribution and shapes of objects or the background, while avoiding minority-class collapse and increasing the likelihood of generating objects during sampling. Simply re-weighting or increasing the model’s update steps for objects cannot achieve the same effect.
> 4. We mainly experimented on SemanticKITTI and CarlaSC to remain consistent with previous work, but we also agree that validating our approach on indoor datasets would be very interesting, haha.
>
> In summary, due to the unsatisfactory score, we will further optimize our work. Please look forward to the next improvements of Comp-D3PM. We wish you all the best.
>
> Sincerely, Authors of Comp-D3PM

---

### Official Review · Reviewer_ARTA · 2025-10-31

**Soundness:** 2
**Presentation:** 1
**Contribution:** 1
**Rating:** 2
**Confidence:** 4

**Summary:**

The paper targets a failure mode in discrete diffusion for 3D semantic scenes where extreme class imbalance causes background to overwhelm foreground (“probabilistic flow collapse”). It proposes a compositional pipeline: (1) split foreground/background into two diffusion streams with distinct transition dynamics, (2) compress features via a BEV-aware DiT block, (3) apply a TF-IDF-like downsampling to reduce background dominance, and (4) fuse the streams during sampling. The method is evaluated on SemanticKITTI and CarlaSC for unconditional scene generation/completion against to 2 baselines.

**Strengths:**

+ Addresses a real, common issue (severe class imbalance) with a simple, implementable recipe.
+ Authors should clear qualitative improvements at object boundaries
+ The idea is clear and easy to follow.

**Weaknesses:**

+ However, the novelty feels incremental: The core idea of splitting a scene into foreground (FG) and background (BG) components and using a compositional approach (generate BG, then generate FG conditioned on BG) is a well-established strategy in generative modeling. This FG/BG separation is a foundational concept, especially in 3D scene understanding where static backgrounds and dynamic foreground objects are often modeled independently. In fact, much of the field is already moving in this direction, with separate research tracks for dynamic 4D object generation and static scene reconstruction. While the authors effectively apply this idea to D3PMs for semantic scenes, the high-level concept itself is not new.

+ Moreover, the architectural choices, such as the BEV-aware DiT (BEDIT), are strong but are primarily effective adaptations of established components (DiT, BEV representations) rather than novel contributions themselves.

+ Related to the first point, the "probabilistic flow collapse" phenomenon is an excellent description of the problem, but it is not "resolved". The paper does not fix this collapse within the diffusion model itself. Instead, it sidesteps the problem with a compositional architecture. This is an effective engineering solution, but it does not represent the fundamental advance as Sec. 3.2 might imply. Actually, long-tail class imbalance is a long standing research area in machine learning, and there are many existing techniques (e.g., re-weighting) that could be adapted to diffusion models to fundementally address this issue without resorting to a compositional pipeline, which add a significant amount of computational overhead.

**Questions:**

I would appreciate it if the authors could clarify the following points:

+ The evaluation is limited to two autonomous driving datasets, SemanticKITTI and CarlaSC. How do the authors expect the Comp-D3PM framework to perform on other types of 3D semantic data, such as indoor scenes (e.g., ScanNet), which have significantly different class distributions, object densities, and structural priors?

+ The proposed method seems to rely on a manually defined, clear-cut separation of classes into "foreground" and "background" (as implied by Fig. 2). How does the model handle ambiguous classes (e.g., "terrain")? Does this hardcoded categorical split limit the model's flexibility, and how would it be adapted to a new dataset with different categories?

+ Could the authors provide a more direct comparison of the computational overhead (e.g., total inference time/steps, FLOPs) of the proposed three-stage process (BG generation, FG generation, fusion) compared to the single-stage baselines?

+ The proposed method only compared with limited baselines with few empirical results, which is a bit lack persuasiveness to me. Could the authors compare their method with more recent diffusion-based 3D scene generation methods to better demonstrate the effectiveness of their method?

---

> ### Author Response · Authors · 2025-11-19
>
> Dear Reviewer,
>
> Thank you for your valuable feedback.
>
> 1. We conducted experiments on the SemanticKITTI and CarlaSC datasets primarily because this is standard practice in previous work. We aimed to maintain consistency with prior studies.
> 2. As described in the paper, we partition objects and background based on the number of voxels per category and whether the category is countable or uncountable. For example, “car” is a countable noun and thus classified as an object, while “terrain” is uncountable and classified as background. This approach is consistent with the Panoptic Segmentation literature. When applying our method to new datasets, we can similarly partition objects and background based on countability.
> 3. Since the models generating objects and background use downsampled BEV representations, the computational complexity is greatly reduced. For example, a 256×256×32 scene is downsampled to a 32×32 representation. The combined time for generating both objects and background is roughly one-third of that required by a full-resolution model, and the overall sampling time is about 4/3 that of a single-stage method.
> 4. SemCity and PDD are the latest unconditioned semantic scene generation methods. Other methods either require GT-compressed BEV representations as conditions or generate full 4D scenes, making direct comparison difficult.
>
> In summary, due to the unsatisfactory score, we will further optimize our work. Please look forward to the next improvements of Comp-D3PM. We wish you all the best.
>
> Sincerely, Authors of Comp-D3PM

---

### Meta-Review · Area_Chair_m7Qs · 2026-01-02

**Summary:**

This paper considers discrete diffusion for 3D semantic scene synthesis under extreme class imbalance and proposes a compositional foreground–background diffusion framework. Reviewers agree that the problem is meaningful and that the method yields improvements on driving datasets. However, the approach is largely based on established foreground–background decomposition and incremental adaptations of existing diffusion and BEV-based components. Despite clarifications and additional experiments, the contribution does not meet the ICLR bar. The AC recommends rejection.

**Reviewer Concerns:**

The main concerns relate to contribution strength and scope. While the paper introduces a clear formulation of the imbalance issue and demonstrates empirical gains, the proposed solution relies on a compositional pipeline rather than a substantive advance in diffusion modeling. Similar decomposition strategies are well known, and the work does not sufficiently differentiate itself from prior approaches. Evaluation is limited to two driving datasets, with incomplete comparisons to stronger baselines and limited evidence of generalization. These concerns remain after the rebuttal.

**Reviewer Scores:**

- Reviewer ARTA: 2, viewing the contribution as incremental and the evaluation insufficient. This assessment would remain unchanged.
- Reviewer fY8p: 4, acknowledging improvements but noting limited generality and missing comparisons. The score would remain below the acceptance threshold.
- Reviewer yPUW: 4, citing issues around methodology and evaluation. The assessment would remain unchanged.
- Reviewer XMZv: 4, raising concerns about the necessity and overhead of the full pipeline. After the rebuttal, the score would likely remain at a similar level.

---

### Decision · Program_Chairs · 2026-01-26

Reject